# Entomopathogenic Potential of *Bacillus velezensis* CE 100 for the Biological Control of Termite Damage in Wooden Architectural Buildings of Korean Cultural Heritage

**DOI:** 10.3390/ijms24098189

**Published:** 2023-05-03

**Authors:** Jae-Hyun Moon, Henry B. Ajuna, Sang-Jae Won, Vantha Choub, Su-In Choi, Ju-Yeol Yun, Won Joung Hwang, Sang Wook Park, Young Sang Ahn

**Affiliations:** 1Department of Forest Resources, College of Agriculture and Life Sciences, Chonnam National University, Gwangju 61186, Republic of Korea; mjh132577@naver.com (J.-H.M.); ajunahenry@mmu.ac.ug (H.B.A.); lazyno@naver.com (S.-J.W.); vanthachoub@gmail.com (V.C.); suin917@naver.com (S.-I.C.); 2671876@naver.com (J.-Y.Y.); 2Forest Products and Industry Department, Wood Industry Division, National Institute of Forest Science, Seoul 02455, Republic of Korea; wonjoung@korea.kr; 3Department of Landscape Architecture, College of Agriculture and Life Sciences, Chonnam National University, Gwangju 61186, Republic of Korea

**Keywords:** wooden buildings, national treasures, termite damage, bacterial entomopathogenic, cuticle disintegration, termiticidal effect

## Abstract

Biocontrol strategies are gaining tremendous attention in insect pest management, such as controlling termite damage, due to the growing awareness of the irreparable harm caused by the continuous use of synthetic pesticides. This study examines the proteolytic and chitinolytic activities of *Bacillus velezensis* CE 100 and its termiticidal effect through cuticle degradation. The proteolytic and chitinolytic activities of *B. velezensis* CE 100 systematically increased with cell growth to the respective peaks of 68.3 and 128.3 units/mL after seven days of inoculation, corresponding with the highest cell growth of 16 × 10^7^ colony-forming units (CFU)/mL. The in vitro termiticidal assay showed that *B. velezensis* CE 100 caused a rapid and high rate of termite mortality, with a median lethal time (LT50) of >1 h and the highest mortality rates of 91.1% and 92.2% recorded at 11 h and 12 h in the bacterial broth culture and crude enzyme fraction, respectively. In addition to broken setae and deformed sockets, termites treated with the bacterial broth culture exhibited degraded epicuticles, while the crude enzyme fraction caused severe disintegration of both the epicuticle and endocuticle. These results indicate the tremendously higher potential of *B. velezensis* CE 100 in the biological control of subterranean termites compared to the previously used entomopathogenic bacteria.

## 1. Introduction

Termites (Isoptera: Rhinotermitidae) are eusocial detritophagous feeders with symbiotic hindgut protozoa that enable them to feed on cellulosic wood matter, causing enormous wood damage, especially in wooden architectural buildings of cultural heritage worldwide [1,2,3,4]. Traditional wooden architectural buildings have been mainly designated as national treasures in various countries due to their significance as rich assets of cultural heritage [2,3]. In Asia, traditional wooden structures are mainly built using pine wood (*Pinus densiflora*), which is highly palatable and, thus, prone to tunneling damage caused by subterranean termite (*Reticulitermes speratus kyushuensis* Morimoto) species [2,5,6]. Generally, wood tunneling damage by insect pests such as subterranean termite species is among the major factors that cause structural deterioration in wooden buildings of cultural heritage [2,3,4,6,7]. In Republic of Korea, such damage is often higher in warmer regions such as the southern provinces Jeonnam and Gyeongnam compared to the northern provinces Gyeonggi and Gangwon, which are often colder [8]. Termite activity is greatly affected by temperature, and new invasive termite species are increasingly being reported in new habitats due to climate change [9,10,11]. Evidently, the rising global temperature is a contributing factor to increased activity and escalating levels of termite damage across the world [8,9,12]. Despite the difficulties in the precise determination of the economic losses caused by termite damage, the available reports point to an undisputedly increasing trend in termite activity (and other tunneling insect pests) due to changes in global climate [9,13]. Besides the irreversible tunneling damage to the appearance and stability of wooden architectural buildings, termite damage also entails the cost of pest control, the side effects of termiticides, the disruption of eco-tourism, interference with social-cultural traditions, and the loss of cultural heritage, which cannot be economically estimated [2,3,14].

There are several effective methods that have been used in the control of termites across the world, but most of these are based on the application of chemical termiticides [14,15]. Several studies have reported termite control strategies, most of which involve the application of synthetic insecticides/termiticides as wood preservatives, termite baits, and fumigants as the most commonly preferred methods [15,16,17]. The application of chitin synthesis inhibitors (CSIs) such as noviflumuron and insect growth regulators (IGRs) such as hexaflumuron insecticides, formulated into durable and palatable cellulosic baits, could eliminate an entire termite colony in a period of approximately 2–3 months [16,18,19,20]. However, to eliminate the colony and, thus, protect wooden structures from attack, the termite feeding activity should remain strictly uninterrupted during a relatively long critical feeding period, lest they abandon the bait for other alternative forages [20]. Colony elimination using the ‘termite bait strategy’ requires clearing the neighboring alternative palatable wood trees around the protected structure so the termites will continuously forage only on the bait for a sufficient period [20,21]. However, alternative palatable forage sources such as *P. densiflora* are always an integral element of the Korean landscape around the premises of wooden architectural buildings of traditional heritage [22]. Therefore, it is unsustainable to clear these precious trees around the protected structures because of their invaluable social-economic significance and microclimate management [22,23]. Thus, controlling termite damage in wooden structures using CSI or IGR termite baits is still unreliable since it would be practically impossible to prevent interruptions (abandoning the bait for alternative forage) during the critical feeding period [20]. In addition, the continuous use of chemical insecticides has been consistently associated with devastating impacts on human life, animals, and non-target organisms as well as the risk of insecticide resistance. This, in turn, compromises environmental health, lowers biodiversity, and increases pest outbreak [17,24,25,26,27].

Among the available alternatives to synthetic chemical-based termiticides, the use of entomopathogenic microbes such as bacteria is increasingly becoming favorable as a safer and effective strategy [28,29]. Specifically, entomopathogens of *Bacillus* spp. have been reported to exert termiticidal effects through, among other factors, their proteolytic and chitinolytic activities [28,30]. The prospect of proteolytic and chitinolytic bacterial entomopathogens as a biocontrol alternative to chemical insecticides is based on the enzymatic degradation of the cuticle layer, which is a vital structural component for the survival of insect pests [31,32,33]. Recently, Moon et al. (2023) reported the termiticidal potential of *Bacillus licheniformis* PR2, which secretes protease and chitinase hydrolytic enzymes, against *R. speratus* worker termites [34]. The enzymes produced by *B. licheniformis* PR2 are known to degrade the cuticle layers in the exoskeleton of insect pests, which leads to lethal effects on subterranean worker termites [32,34]. The faster degradation rate of the cuticular glycoproteins and chin polymers by bacterial proteases and chitinase enzymes could be vital to achieving rapid termite mortality with less chance of survival through escaping termites. Once the cuticle is degraded, termite death is inevitable due to their extremely low tolerance to desiccation [35,36]. A previous study by Moon et al. (2023) reported a termiticidal effect of entomopathogenic *B. licheniformis* PR2 on subterranean termites. However, the efficacy of the entomopathogen was slow in the first five hours after treatment, which could lead to termite escape in a natural environment. Therefore, there is a need to explore alternative bacterial agents that could offer a more rapid and effective termiticidal effect. The objective of this study was to investigate the termiticidal potential of the *B. velezensis* CE 100 entomopathogen, which has prolific proteolytic and chitinolytic activity, in the biocontrol of *R. speratus* worker termite pests.

## 2. Results

### 2.1. Cell Growth and Protease and Chitinase Activity of Bacillus velezensis CE 100

The cell growth of *B. velezensis* CE 100 sharply increased on the first day after inoculation to 5.7 × 10^7^ CFU/mL (Figure 1A). From the second to the sixth day after inoculation, the cell growth of *B. velezensis* CE 100 was relatively gradual, but showed a sharp increase to a peak of 1.7 × 10^8^ CFU/mL on the seventh day after inoculation. A relatively similar pattern was observed for protease and chitinase activity in the bacterial broth culture. Protease activity showed a rather steady pattern of increase from the start up to a peak of 68.3 units/mL on the seventh day after inoculation (Figure 1B). The activity of chitinase was even more consistent with the pattern of cell growth, showing a gradual increase from the start up to 79.0 units/mL on the sixth day, before sharply increasing to a peak of 128.3 units/mL on the seventh day after inoculation (Figure 1C).

After attaining the maximum cell growth, *B. velezensis* CE 100 showed a rapid decline in the number of viable cells to 9.3 × 10^7^ CFU/mL on the eighth day, and remained relatively stable until the end of the experiment. Similarly, the enzyme activity for both protease and chitinase showed a respective decline to 46.3 units/mL and 77.3 units/mL on the ninth day after inoculation, and remained stable until the end of the study. The decline in enzyme activity was rather more gradual than the observed decline in number of viable cells of *B. velezensis* CE 100.

### 2.2. Termiticidal Activity of Bacillus velezensis CE 100 against Reticulitermes speratus Worker Termites

The treatment of worker termites with *B. velezensis* CE 100 bacterial broth culture and crude enzyme fraction caused a termite mortality rate of 62.2% and 70.0% within an hour after treatment (Figure 2). The lethal time 50 (LT50) was >1 h in both the bacterial broth culture and crude enzyme treatment. The termiticidal effect of *B. velezensis* CE 100 increased with time after treatment application, with a significantly higher mortality rate in the crude enzyme treatment compared to bacterial broth culture treatment up to 8 h. The maximum termiticidal activities of *B. velezensis* CE 100 in the broth culture treatment (91.1%) and crude enzyme fraction (92.2%) were recorded after 12 h and 11 h, respectively. There was no significant termite mortality observed in either the control or PB medium groups throughout the 12 h study period (Figure 2).

### 2.3. Morphological Deformation of Termite Cuticles after Treatment

The scanning electron microscope (SEM), GeminiSEM 500 (Carl Zeiss AG, Oberkochen, Germany) examination of termite cuticles from the different treatment groups revealed striking differences that could demonstrate the mode of termiticidal effect of *B. velezensis* CE 100. The cuticle morphologies of termites in both the control group and PB medium were healthy, with intact setae, and no visible laceration or rapturing were observed (Figure 3A,B).

However, the cuticles of termites treated with either the broth culture or crude enzymes of *B. velezensis* CE 100 exhibited deleterious changes including broken setae, ruptures, and severe deformations in the cuticle morphology (Figure 3C,D). Specifically, the termites treated with *B. velezensis* CE 100 broth culture showed ruptured epicuticles, some broken setae, and swollen sockets (Figure 3C), while the termites treated with the crude enzyme fraction of *B. velezensis* CE 100 exhibited a more severe disintegration of both the epicuticle and endocuticle, with broken setae and severely deformed sockets (Figure 3D).

## 3. Discussion

Based on previous reports, proteolytic and chitinolytic entomopathogens have tremendous potential in the biological control of insect pests through enzymatic cuticle degradation [31,32,33,34,37]. In particular, the termite cuticles, which are often soft and unsclerotized, are less resistant to the hydrolytic activity of cuticle-degrading enzymes and more prone to desiccation, especially if the cuticle is damaged [35,38]. The results of this study indicate that the maximum protease and chitinase enzyme activities from *B. velezensis* CE 100 were 68.3 and 128.3 units/mL, respectively (Figure 1). Consequently, treatment with the bacterial broth culture and crude enzyme fraction of *B. velezensis* CE 100 caused a respective termite mortality of up to 91.1% and 92.2%, with a significantly (*p* < 0.01) higher termiticidal effect in the latter during the first 8 h after treatment (Figure 2). The higher mortality in the crude enzyme fraction further indicates that the production of cuticle-degrading enzymes is the mode of termiticidal effect of *B. velezensis* CE 100, since the partial purification of the crude enzymes improves the insecticidal efficacy [31].

The SEM examination of cuticle deformations in the different treatment groups further revealed that the crude enzyme fraction of *B. velezensis* CE 100 caused more severe cuticle disintegration, with complete rupturing of both the epicuticle and procuticle, compared to the bacterial broth culture, where visible deformations were only observed in the epicuticle (Figure 3C,D). The simultaneously dual enzymatic effect of protease and chitinase could also have enhanced the degradation/hydrolysis on termite cuticles [33]. This could be explained by the fact that the degradation of the protein fibers that embed the chitin polymers increases the exposure of the substrate/surface area for chitinase activity [31,33]. Leger et al. (1986) confirmed this phenomenon by demonstrating an increase in the concentration of N-acetyl d-glucosamine monomers from chitin hydrolysis by 1.5-fold when protease and chitinase enzymes were applied together compared to when chitinase was applied alone [33]. However, in this study, both the control and PB medium groups caused neither any significant termite mortality nor any cuticle deformation symptoms during the 12 h experimental period (Figure 3A,B). Thus, the termite mortality and the degradation of termite cuticles were a direct result of the proteolytic and chitinolytic activity of *B. velezensis* CE 100. The termite cuticle, like in other insects, is mainly composed of structural proteins such as collagen and some functional globular proteins [39,40]. This cuticular protein matrix can be diversely modified, embedded into chitin polysaccharide, and interlinked to sugar residues by hydrogen bonds to form strong structures with various properties and functions [39,41,42]. Thus, the degradation of cuticular protein fibers and chitin polymers by protease and chitinase enzymes produced by bacterial entomopathogens causes the disintegration of an insect’s cuticle, which is a vital structural and functional organ in the insect’s life [31,32,33,34,37]. Cuticle disintegration causes loss of moisture and leads to a rapid rate of termite mortality due to poor desiccation tolerance [35,36].

In a previous study, the proteolytic and chitinolytic activity of *B. licheniformis* PR2 was also considered to be the major cause of termite mortality [34]. By contrast, *B. velezensis* CE 100 resulted in a notably faster and higher termiticidal effect than the previously reported rates using *B. licheniformis* PR2 under similar experimental conditions. For instance, the median lethal time (LT50, which is the time required to achieve 50% mortality after treatment application) in subterranean worker termites was reduced to less than 1 h when treated with *B. velezensis* CE 100, compared to at least 5 h observed with *B. licheniformis* PR2 in the previous study [34]. This rapid termiticidal efficacy of *B. velezensis* CE 100 is vital for minimizing the chances of escape if the treatments are applied in vivo. The rapid/faster rate of termite mortality caused by *B. velezensis* CE 100 is also consistent with the higher proteolytic and chitinolytic activity (Figure 1B,C) compared to previously reported values using *B. licheniformis* PR2 [34]. The maximum activity of protease and chitinase enzymes in *B. licheniformis* PR2 under similar conditions was previously reported to be 35.9 units/mL and 82.3 units/mL, respectively [34]. Thus, the proteolytic and chitinolytic activity of *B. velezensis* CE 100 recorded in this study was 1.9- and 1.6-fold higher than that of *B. licheniformis* PR2 reported in the previous study. Due to such substantially higher proteolytic and chitinolytic activity, *B. velezensis* CE 100 is indeed expected to exhibit a notably more efficient termiticidal activity under similar conditions. Moreover, a comparison of the two entomopathogens reveals deeper cuticle rupturing and setae deformations when termites were treated with *B. velezensis* CE 100 compared to the previously reported termiticidal symptoms of *B. licheniformis* PR2 treatment [34]. Due to its rapid termiticidal efficacy, *B. velezensis* CE 100 could be considered as a more promising entomopathogen in the biological control of subterranean termites. *B. velezensis* CE 100 was also previously reported to effectively control jujube gall midges (*Dasineura jujubifolia*) in the laboratory and in field conditions [31]. Therefore, based on the insecticidal versatility coupled with such rapid and high termiticidal activity, *B. velezensis* CE 100 could be a suitable biopesticide candidate. This versatility and high efficacy are a direct attribute to its prolific production of protease and chitinase enzymes, which degrade the cuticular protein fibers and chitin polysaccharides, which are integral components of the insects’ exoskeleton.

## 4. Materials and Methods

### 4.1. Preparation of Bacterial Broth Culture and Termite Samples

The entomopathogenic strain *B. velezensis* CE 100 was isolated from pot soil grown with tomato plants and tested for nematocidal activity as previously described [43]. The stock culture of *B. velezensis* CE 100 (10^7^ colony-forming units (CFU)/mL) was obtained from MC Biotech Co. Ltd. (Jeonnam-do, Republic of Korea) and preserved in 50% glycerin at −80 °C for use in further experiments. To obtain the culture for use in this study, the bacterial strain was spread on tryptone soy agar (TSA) medium and incubated at 30 °C for 24 h. To avoid dissociation, a single colony of the bacterium was picked, pre-inoculated in 200 mL of pink broth (PB) medium, and incubated in a shaking incubator (H1012, Benchmark Scientific Inc., Edison, NJ, USA) set at 30 °C and 120 rpm for 48 h, and this culture was used in all the subsequent experiments in this study [44]. The PB medium was prepared by mixing 0.3% chitin powder, 1% colloidal chitin, 0.3% sucrose (as carbon source), 0.3% N.P.K ((20:20:20), 0.3% yeast extract, and mineral salts 0.02% KH_2_PO_4_, 0.02% MgSO_4_, 0.01% CaCO_3_, and 0.01% NaCl.

The termite samples were collected from infested poles of *P. densiflora* of a Korean traditional wooden architectural house in Gwangju-si, Republic of Korea (Figure 4A). These palatable poles of *P. densiflora* attract the foraging *R. speratus* worker termites as previously discussed [34]. One of the infested poles was carefully split open, and a section with a sufficient quantity of actively foraging worker termites was wrapped in plastic film to prevent moisture loss and quickly transported to the laboratory to be used in the termiticidal assay.

### 4.2. Cell Growth, Proteolytic Activity, and Chitinolytic Activity of B. velezensis CE 100

The growth pattern and the activity of protease and chitinase enzymes in the bacterial culture of *B. velezensis* CE 100 were determined from the bacterial broth using PB medium. Briefly, 1 mL of *B. velezensis* CE 100 pre-inoculum described above was used for inoculating 1 L of the medium, and the bacterial culture was grown at 30 °C and 120 rpm in a shaking incubator. During growth, 1 mL, 2 mL, and 2 mL samples of the bacterial culture were separately collected on a daily basis for ten days for the determination of cell growth curve and protease and chitinase activities, respectively. For the determination of cell growth pattern, 10-fold serial dilutions of the samples were performed under sterile conditions, and 100 µL of each sample was then spread on TSA plates. The plates were subsequently incubated at 30 °C after 24 h to enumerate the viable cells of *B. velezensis* CE 100.

For the protease enzyme activity, the method previously described by Moon et al. (2021) was used, with slight modifications [45]. Briefly, 100 mM of tris buffer (pH 8.0) was prepared with 2 mM CaCl_2_ and 1% casein on the day of use. Then, 2 mL of the bacterial samples collected above was centrifuged at 12,000 rpm for 10 min to collect the supernatant. Then, 50 µL of the supernatant was mixed with 950 µL tris buffer and incubated at 60 °C for 15 min, and the reaction was terminated by adding 500 µL of 20% trichloroacetic acid. After centrifugation at 13,000 rpm for 15 min, the absorbance of acid-soluble proteins in the supernatant was measured using a UV spectrophotometer (UV-1650PC, Shimadzu, Kyoto, Japan) at 280 nm. A unit of protease activity was defined as the amount of enzyme required to liberate 1 µg of tyrosine per minute from the hydrolysis of the casein substrate.

The activity of the chitinase enzyme was determined following the method described by Hong at al. (2022), based on the hydrolysis of chitin substrate [46]. Briefly, 0.5% of colloidal chitin substrate and 50 mM of sodium acetate buffer adjusted to pH 5.0 were separately prepared. Then, 2 mL of the bacterial samples collected on a daily basis (as described above) was centrifuged at 12,000 rpm for 10 min. Then, 50 µL supernatants from each sample was homogenized in 500 µL of colloidal chitin, 450 µL of sodium acetate buffer was added, and the mixture was incubated for 1 h at 37 °C. After incubation, 200 µL of NaOH (1N) solution was added to the mixture and vortexed to terminate the reaction. The mixture was centrifuged for 10 min at 12,000 rpm, 750 µL of the supernatant was mixed with 250 µL of de-ionized water, 1 mL of Schales reagent was added, and the mixture was boiled at 100 °C for 15 min for color development. The absorbance was then measured on a UV spectrophotometer at a wavelength of 420 nm, and the concentration of N-acetyl-glucosamine (the product of chitin hydrolysis) was calculated from the standard curve (unit/mL). A unit of chitinase activity was defined as the amount of enzyme required to liberate 1 µmol of N-acetyl-glucosamine/hour from the hydrolysis of chitin at 37 °C. Each experiment (cell growth curve, protease, and chitinase assay) was repeated three times, and the average values were used.

### 4.3. Preparing the Crude Enzyme Fraction from B. velezensis CE 100 Bacterial Broth Culture

The crude enzyme fraction was prepared as described by Choub et al. (2021) from the 7-day bacterial broth culture of *B. velezensis* CE 100 grown in PB medium at 30 °C and 130 rpm [44]. The cells were separated through centrifugation at 6000 rpm and 4 °C for 30 min, and the supernatant was filtered through four layers of filter paper (Whatman No. 6, Whatman International Ltd., Maidstone, UK). The filtrate was gradually precipitated by adding ammonium sulfate salt up to 80% saturation, with gentle stirring using a magnetic stirrer. The flask containing the filtrate was placed in an ice box (on top of the magnetic stirrer), and melted ice was frequently replaced throughout the protein precipitation process to keep the temperature below 4 °C. The saturated solution was further stabilized by allowing it to stand at 4 °C overnight in a refrigerator, and the precipitate was separated through centrifugation at 6000 rpm for 30 min. The protein pellet was then dissolved in a minimal amount of Tris-HCl buffer (20 mM, adjusted to a pH of 8.2), and the solution was dialyzed at 4 °C for 24 h against a similar buffer solution (the buffer was replaced three times during dialysis after observing color change). The crude enzyme fraction was then kept at −70 °C until use in further experiments.

### 4.4. Termiticidal Activity of B. licheniformis PR2 Broth Culture and Crude Enzymes against R. speratus Worker Termites

The termiticidal experiments were conducted in insect growth chambers (VS-91G09M-1300, VISION Co., Daejeon, Republic of Korea) with relative humidity (RH) set at 70 ± 5% and temperature at 25 ± 2 °C, and the lighting conditions were achieved using a piece of black cotton. The experimental groups were as follows: (1) control (termites treated with only sterile distilled water), (2) PB medium (termites treated with only PB medium without *B. licheniformis* PR2 inoculation), (3) bacterial broth culture (termites treated with *B. velezensis* CE 100 broth culture), and (4) bacterial crude enzyme fraction (termites treated with the crude enzyme fraction of *B. velezensis* CE 100). For each replicate, ten worker termites were placed on a piece of sterile wet filter paper placed inside well-ventilated polystyrene insect rearing dishes, measuring 100 mm in diameter and 40 mm in height with a ventilation hole size of 40 mm, covered with a mesh of size 0.053 mm (SPL Life Science Co., Ltd., Pocheon-si, Republic of Korea). The termites were sprayed with 1 mL of each treatment solution, and the experiment was conducted at 25 ± 2 °C 70 ± 5% RH, in the insect growth chamber. The mortality rate for *R. speratus* worker termites was evaluated hourly for 12 h after treatment and the experiment was repeated three times. Termite mortality, M (%), was calculated as M (%) = 100 × (D/I), where D is the number of dead termites at a given time, and I is the initial number of termites in each treatment group. Dead termites were confirmed when they did not respond after they were touched with a soft brush while observing their response on an optical microscope (BX41TF Microscope, Olympus, Tokyo, Japan) at 10 × magnification. The dead termites were separately collected according to each treatment and fixed in 4% paraformaldehyde every hour. After 12 h, the surviving termites were also fixed, and all samples were stored at 4 °C in sterile vials for further microscopic analysis.

### 4.5. Analysis of the Morphological Changes in Termite Cuticles after Treatment

The examination of termite cuticle morphologies under different treatment conditions was carried out as described by Choi et al. (2022) using an SEM [31]. The termite samples were fixed for at least 48 h in 4% paraformaldehyde and then washed three times in phosphate-buffered saline (PBS, at a pH of 7.4). Then, the samples were dehydrated in a series of 40, 60, 80, 95, and 100% at an interval of 30 min/step using ethanol as a dehydrating agent. After ethanol dehydration, the samples were further dehydrated in a series of isoamyl acetate/ethanol (*v*/*v*) ratios of 1:2 for 20 min, 1:1 for 30 min, and pure isoamyl acetate for 1 h. After double dehydration, the samples were air-dried under a fume hood for 24 h. Then, the samples were mounted on aluminum pin stubs using carbon conductive tapes, followed by gold coating at 60 °C. The SEM imaging of termite cuticles was carried out at an acceleration voltage of 15 kV and a magnification of 500×.

### 4.6. Statistical Analysis

Statistical analysis of all the data reported in this study was performed using SPSS 25.0 statistical software (SPSS Inc., Chicago, IL, USA). The cell growth and enzyme activity data were subjected to analysis of variance (ANOVA) using the Waller–Duncan test, while termite mortality data were subjected to two-way ANOVA at *α* = 0.01. All the data are presented as mean ± standard deviation of three replications (*n* = 3).

## 5. Conclusions

In the search to find more options for the eco-friendly management of insect pests, screening for new and more effective entomopathogenic bacteria as alternatives to synthetic insecticides is becoming important. The use of entomopathogenic bacteria such as *B. velezensis* CE 100 in the control of termite pests is of profound importance as an alternative or supplementary measure to existing chemical-based strategies in combating termite damage, especially in wooden architectural buildings of cultural heritage. The results of this study demonstrate that *B. velezensis* CE 100 has prolific proteolytic and chitinolytic potential, which is directly related to bacterial cell growth. The treatment of termites with *B. velezensis* CE 100 causes rapid degradation of the soft unsclerotized termite cuticles, which leads to desiccation and invasion of toxic substances to the inner body tissues, resulting in termite mortality. A solid-state bacterial culture of *B. velezensis* CE 100 could be applied to the wood material used for the construction of wooden architectural buildings in cultural heritage sites for eco-friendly termite control. The bacterial broth culture could also be periodically sprayed around wooden buildings to prevent termites from invading the soils and wood habitats around the protected structures or applied to eliminate the termite colony.

## Figures and Tables

**Figure 1 ijms-24-08189-f001:**
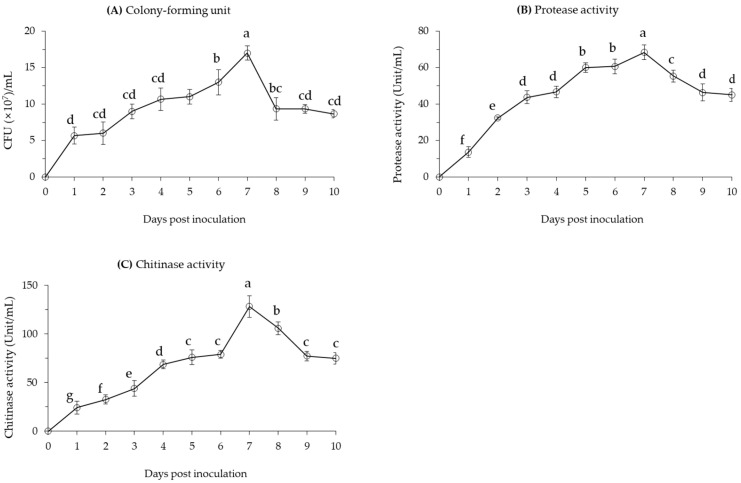
(**A**) Cell growth, (**B**) protease enzyme activity, and (**C**) chitinase enzyme activity of *B. velezensis* CE 100 in PB medium at 30 °C for ten days. All points represent mean ± standard deviation (*n* = 3). Different superscripts in a figure indicate that means are significantly different (*p* = 0.01).

**Figure 2 ijms-24-08189-f002:**
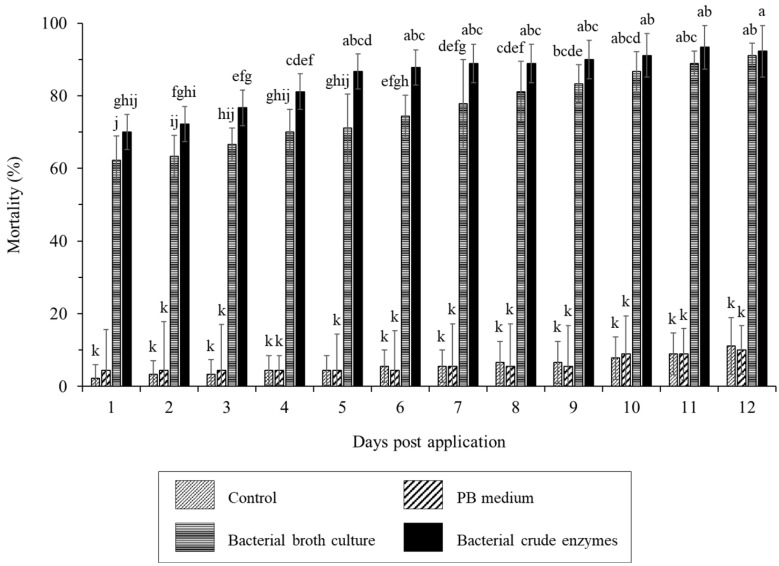
The mortality rate of *R. speratus* worker termites in the control group, PB medium, bacterial broth culture, and bacterial crude enzyme fraction from *B. velezensis* CE 100 under laboratory conditions. The bars represent mean ± standard deviation (*n* = 3). Different superscripts a–j in the figure indicate that values are significantly different (*p* = 0.01).

**Figure 3 ijms-24-08189-f003:**
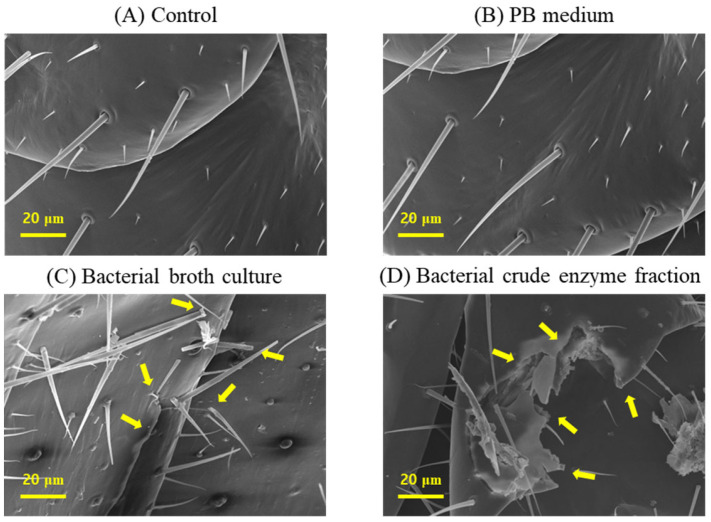
Micrographs of termite cuticles from a scanning electron microscope, showing changes in the cuticle morphologies of termites in different treatment groups: (**A**) control, (**B**) PB medium, (**C**) bacterial broth culture, and (**D**) bacterial crude enzyme fraction, from *B. velezensis* CE 100. Arrows indicate cuticle deformations.

**Figure 4 ijms-24-08189-f004:**
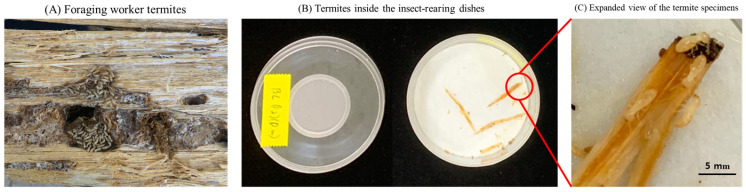
(**A**) Foraging worker termites inside a split piece of wood obtained from a traditional Korean house, (**B**) experimental termite specimen collected on a moist piece of filter paper inside an insect rearing dish, and (**C**) an expanded view of the termite specimens inside insect rearing dish before treatment.

## Data Availability

All the data are available on request from the corresponding author.

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
