# Peer review of "Entomopathogenic Potential of Bacillus velezensis CE 100 for the Biological Control of Termite Damage in Wooden Architectural Buildings of Korean Cultural Heritage"

_ijms, 2023, doi:10.3390/ijms24098189_

Round 1

Reviewer 1 Report

The paper is well written and an interesting study.

In the introduction of the abstract the authors miss to write that the study was on termites.

line 46 ... in Korea? or Asia?

At the end of the Introduction the authors write about B. licheniformis PR2 and B. velezensis CE100, giving the impression the both entomopatogens will be compared. As this was not done (but would be very interesting) this should be stated clearyl.

I dont have any further comments on the paper, well written.

Reviewer 2 Report

Dear authors,
Do we know why termites allow termite infection? In the case of fungi, they can detect the smell of fungal spores, and already at the entrance the guards take the infected worker aside, cut off her head and then commit suicide themselves. In this way they protect other individuals, the larvae and especially the queen. When Paul Stamets found a way to delay the spores of entomopathogenic fungi, it was the sentinels who let the infected workers inside the colony, whereupon they died. Could there be a similar mechanism with Bacillus? Is a single application of the bacterium enough to wipe out an entire colony? Do infected individuals spread the bacteria in the tunnels and infect other insects through contact?
What is the risk that the introduced bacterium will not also destroy other beneficial insects and reduce biodiversity? If it is not selective, can we speak of environmentally friendly management of harmful insects? The worst would be if the bacterium attacked pollinators. Have experiments been carried out with bees, for example? The application in residential buildings could also have a positive effect on reducing the population of other insect pests such as ants or cockroaches.
